# Cushing’s Disease: Long-Term Effectiveness and Safety of Osilodrostat in a Polish Group of Patients with Persistent Hypercortisolemia in the Experience of a Single Center

**DOI:** 10.3390/biomedicines11123227

**Published:** 2023-12-06

**Authors:** Lukasz Dzialach, Joanna Sobolewska, Wioleta Respondek, Katarzyna Szamotulska, Przemysław Witek

**Affiliations:** 1Department of Internal Medicine Endocrinology and Diabetes, Medical University of Warsaw, 03-242 Warsaw, Polandprzemyslaw.witek@wum.edu.pl (P.W.); 2Department of Internal Medicine Endocrinology and Diabetes, Mazovian Brodnowski Hospital, 03-242 Warsaw, Poland; 3Department of Epidemiology and Biostatistics, Institute of Mother and Child, 01-211 Warsaw, Poland; katarzyna.szamotulska@imid.med.pl

**Keywords:** adrenal steroidogenesis inhibitors, cortisol, Cushing’s disease, Cushing’s syndrome, medical therapy, osilodrostat

## Abstract

Osilodrostat is a potent oral steroidogenesis inhibitor that has emerged as the new medical agent for patients with Cushing’s disease (CD) requiring long-term medical therapy for hypercortisolemia control. Its efficacy and safety have been assessed in clinical trials; however, real-world evidence is still scarce. This study aimed to investigate the long-term treatment (156 weeks) clinical and biochemical effect of osilodrostat in six patients with CD at a single center in Poland, initially participating in the LINC4 study. At week 36, all six patients met the key secondary endpoint of the LINC4 trial, achieving normalization of median urinary free cortisol. Osilodrostat treatment allowed for complete disease control in all patients and none of the patients was excluded due to the lack of treatment effectiveness in 156 weeks of follow-up. All patients demonstrated significant improvement from baseline on most metabolic and cardiovascular parameters, which was most evident at week 36 and sustained throughout the study period. This study supports and strengthens the role of osilodrostat as an effective long-term medical treatment in patients with CD. We also present three patient case histories in detail to highlight the clinical situations that endocrinologists might face during osilodrostat therapy.

## 1. Introduction

Endogenous Cushing’s syndrome (CS) is a very rare endocrine condition with an estimated annual incidence of 0.2–5 per million per year. The most common cause of CS, accounting for approximately 70% of all cases, is an adrenocorticotropin (ACTH)-secreting pituitary neuroendocrine tumor (Pit-NET), traditionally defined as Cushing’s disease (CD) [1]. Sustained hypercortisolemia is linked with significantly impaired quality of life, morbidity, and mortality [2,3,4,5,6,7,8]; therefore, normalization of cortisol overproduction while avoiding permanent hormone deficiency and drug dependence is the ideal goal of CD treatment. While the therapy of choice for CD is transsphenoidal surgery (TSS), some patients require additional treatment, including pharmacological agents, radiotherapy, radiosurgery or bilateral adrenalectomy [9,10]. Pharmacotherapy is especially indicated when neurosurgery is unsuccessful, contraindicated, not feasible or not accepted, as well as in recurrent cases and as bridging therapy, while awaiting other procedures [9,10,11]. Medical treatment options include inhibitors of steroidogenesis, pituitary-targeted drugs, and glucocorticoid receptor antagonists [10,12,13,14].

Osilodrostat is a particularly potent oral inhibitor of 11β-hydroxylase (CYP11B1), which catalyzes the final step of cortisol synthesis and represents a novel approach to achieving eucortisolism in patients with CD [15,16]. In the phase III LINC3 study (NCT02180217), evaluating the efficacy and safety of osilodrostat, normalization of mean urinary free cortisol (mUFC) was achieved in 66.4% of patients at the end of the trial [17]. In the phase III LINC4 study (NCT02697734), with an initial placebo-controlled phase, osilodrostat has been shown to be significantly superior to placebo at normalizing mUFC at week 12 (77% vs. 8%) and led to sustained mUFC levels below the upper limit of normal (ULN) in 68.5% of patients in 48 weeks of observation [18]. In the recently published results from the LINC4 study extension (NCT02180217), 72.4% of patients maintained mUFC values below the upper limit of normal (ULN) at their extension end-of-treatment visit (up to 96 weeks) [19].

Data from the clinical practice on patients with CD treated with osilodrostat are limited. Hence, this study aimed to present the clinical and biochemical response of long-term (156 weeks) treatment with osilodrostat in six patients with persistent or recurrent CD at a single center in Poland, with the results of three patients depicted in detail. In contrast to the large clinical studies, our paper is mainly focused on presenting our center’s experience of CD treatment with osilodrostat in real-life settings.

## 2. Materials and Methods

### 2.1. Study Design

The study group included 6 adult patients (4 women and 2 men, aged 35.7 ± 12.8 years) with persistent or recurrent CD (all patients underwent at least one TSS) with uncontrolled hypercortisolemia and who were not suitable for re-operation, enrolled in the phase III LINC4 study at the Department of Internal Medicine, Endocrinology and Diabetes, Medical University of Warsaw, Poland between September 2018 and January 2019. Patients’ baseline characteristics are summarized in Table 1.

Patients must have had an active disease evidenced by mUFC of three 24 h UFC samples above 1.3× the ULN, morning plasma ACTH concentration above or within the reference range (WRR), and a confirmed pituitary source of ACTH excess. The LINC 4 study protocol comprised an initial, randomized, double-blind period (12 weeks) to compare the efficacy of osilodrostat against a placebo and a subsequent 36-week, open-label period complemented by an optional extension (48 weeks) to evaluate the sustained effect of osilodrostat and long-term safety. During the first 12 weeks, patients were randomized in a double-blinded manner to receive osilodorostat 2 mg twice a day (BID) or a matching placebo in accordance with the study protocol. In that phase, dose adjustments were made based on mUFC, rate of mUFC decrease, and drug tolerability. The dose could be increased approximately every 3 weeks with an escalation sequence of 2–5–10–20 mg BID. All 6 patients restarted the open-label period (weeks 13–48) on osilodrostat 2 mg BID and then continued treatment as a part of the extension phase (weeks 48–96). The investigators determined dose adjustments during open-label and extension periods based on mUFC, mean late-night salivary cortisol (mLNSC), clinical presentation, and drug tolerability. The maximum dose of osilodrostat in double-blinded and open-label/extension periods was 20 and 30 mg BID, respectively. The primary endpoint of the LINC4 study was to determine whether osilodrostat was superior to placebo in normalizing mUFC at week 12. The key secondary endpoint was the proportion of patients achieving mUFC normalization at week 36 (after 24 weeks of open-label treatment). Complete biochemical response was defined as mUFC < ULN, and partial biochemical response as mUFC > ULN but ≥50% reduction from baseline. A detailed description of the study has already been presented [18]. After completing their participation in the clinical trial, all patients continued treatment with osilodrostat financed by the National Health Fund.

### 2.2. Clinical Evaluation

The clinical evaluation, performed at baseline and every study visit, included assessment of physical features of CD, signs and symptoms of adrenal insufficiency (AI) and the measurement of weight, body mass index (BMI), waist circumference, blood pressure (BP), and heart rate by standard methods. Electrocardiogram (ECG) with calculated corrected QT (QTcF) intervals was performed pre-dose and 1.5 h post-dose at each visit.

### 2.3. Laboratory Assessment

After the patients’ enrollment in the LINC4 trial, biochemical and hormonal parameters were assessed by the central laboratory designated by the study protocol.

UFC (RR: 11.0–138 nmol/24 h), LNSC (RR: 10–11 PM: ≤2.5 nmol/L), and total serum cortisol (TSC, RR: 8–10 AM: 127–567 nmol/L) were measured by liquid chromatography-tandem mass spectrometry and plasma ACTH by immunoassay (RR: 7–10 AM: 1.3–11.1 pmol/L). mUFC was calculated from two or three 24 h UFC measurements and mLNSC from two LNSC measurements, collected on three or two consecutive days before visits. Metabolic factors, including fasting plasma glucose (FPG), lipids, glycated hemoglobin (HbA1c) and liver parameters (alanine transaminase (ALT) and aspartate transaminase (AST)) were measured by standard methods. After completing the clinical trial, biochemical and hormonal assessments were performed in the laboratory of the Endocrinology Department of the Medical University of Warsaw.

### 2.4. Tumor Imaging

Pituitary imaging was performed using a 1.5-T magnetic resonance imaging (MRI) with gadolinium enhancement with 2.5 mm slice thickness at baseline, weeks 26, 48, 72 and 96, and 144.

### 2.5. Statistic Analysis

Data are presented as mean and standard deviation (SD) or median and range. The Wilcoxon signed-rank test was used for the comparisons between baseline and particular points of observation. All calculations were completed using IBM^®^ SPSS^®^ Statistics 25. The *p*-values < 0.05 were considered as statistically significant.

## 3. Results

### 3.1. Effectiveness and Hormonal Parameters Analysis

At the baseline, in the six patients included in the study, median mUFC and mLNSC values were 480.0 nmol/24 h (3.48× ULN) and 11.40 nmol/L (4.57× ULN), respectively. At week 36 (after 24 weeks of the open-label phase), all six patients met the key secondary endpoint of the LINC4 trial, achieving a mUFC < ULN, at a median osilodrostat dose of 4 mg BID (range: 1–20 mg BID). At that time, the median mUFC was 75.15 nmol/24 h (WRR, range: 34.97–94.60 nmol/24 h; *p* = 0.031) and the median mLNSC was 2.11 nmol/L (WRR, range: 0.8–5.35 nmol/L; *p* = 0.031); four (66.7%) of the analyzed patients had both mUFC and mLNSC < ULN. Median time and osilodrostat dose to first mUFC < ULN (from study restart at week 12) was 5 weeks (range: 2–20 weeks) and 5 mg BID (range: 2–20 mg BID), respectively. Median time and osilodrostat dose to first mUFC and mLNSC < ULN (from study restart at week 12) was 9.5 weeks (range: 2–48 weeks) and 5 mg BID (range: 2–25 mg BID), respectively. The minimum and maximum osilodrostat dose required, at least temporarily, to achieve and maintain eucortisolemia during the whole observation period was 1 mg every 3 days (patient 4) and 30 mg BID (patient 1 and 2), respectively. At 96 weeks’ follow-up, the median mLNSC (1.58 nmol/L, WRR) significantly decreased by 82.75% (*p* = 0.031) compared to baseline (11.43 nmol/L, 4.57x ULN); four of the six patients had an mLNSC < ULN at that time. At 156 weeks’ follow-up, the median mUFC (50.83 nmol/24 h, WRR, range: 17.2-110,6 nmol/24 h) decreased by 92.99% (*p* = 0.031) compared to baseline (480.0 nmol/24 h, 3.48x ULN, range: 220.53–3316.40 nmol/24 h), and morning TSC (281.50 nmol/L, WRR, range: 116.0–420.0 nmol/L) decreased by 33.64% (*p* = 0.031) compared to baseline (448.5 nmol/L, WRR, range: 290–420 nmol/L). However, the median morning TSC was within the reference range through the whole study period. All six patients achieved and maintained a complete treatment response, and none of the patients was excluded due to the lack of treatment effectiveness in 156 weeks of follow-up.

The median ACTH concentration significantly increased during the whole study period, and at 156 weeks’ follow-up (91.6 pmol/L, 8.25x ULN) it was 9.5-fold higher (*p* = 0.031) compared to baseline. Data are summarized in Table 2. Figure 1 presents mUFC, Figure 2 mLNSC, and Figure 3 ACTH evolution according to osilodrostat dose during the study observation for every patient. Figure 4 presents median mUFC (a) and median mLNSC (b) evolution according to median osilodrostat dose during the study observation of all patients.

### 3.2. Effect on Metabolic, Cardiovascular and Liver Parameters

At the baseline, mean (SD) weight, BMI, and waist circumference measurements of the analyzed group were 69.88 (12.34) kg, 26.74 (3.11) kg/m^2^, and 93.67 cm (3.59), respectively. BMI was WRR (18.5–24.9 kg/m^2^) in 1/6 patients, 4/6 patients were overweight (BMI: 25–29.9 kg/m^2^), and 1/6 patients had obesity (BMI: ≥30 kg/m^2^). During observation, weight, BMI, and waist circumference gradually decreased in all patients; mean (SD) weight, BMI, and waist circumference at the end of the observation were 63.75 (11.79) kg, 24.41 (3.13) kg/m^2^, and 84.83 cm (9.06), respectively. BMI was WRR in 4/6 patients, and 2/6 were overweight. The fastest decrease in body weight, BMI, and waist circumference was observed during the first 36 weeks; however, the lowest values were achieved during the last follow-up. One patient, after initial improvement, had a significant increase in body weight between weeks 60 and 96 of the study, which was associated with primary disease progression. Mean (SD) TC decreased from 5.58 (1.05) mmol/L at baseline to 4.02 mmol/L (0.69) at week 96; mean (SD) TAG decreased from 1.94 (1.21) mmol/L at baseline to 1.55 (0.86) at week 96. One of the three patients taking antilipidemic drugs at baseline discontinued the treatment. At the baseline, 2/6 patients were classified as diabetic, 2/6 had IGT, and 1/6 had IFG. Mean (SD) FPG decreased from 5.12 (1.23) mmol/L at baseline to 4.43 mmol/L (0.43) at week 96; mean (SD) HbA1c decreased from 5.5% (0.61) at baseline to 5.35% (0.56) at week 156. The fastest improvement in metabolic parameters was observed during the first 36 weeks; however, the lowest values were achieved during the last follow-up, excluding TG (week 72) and HbA1c% (week 132).

At the baseline, mean (SD) SBP and DBP were 125.0 (8.25) mmHg and 79.23 (7.20) mmHg, respectively. Five of the six patients had hypertension prior to the study and were taking antihypertensive drugs. At the end of the observation, mean (SD) SBP and DBP were 121.17 (3.71) mmHg and 83.33 (7.09) mmHg, respectively. However, considering 5/6 patients were under antihypertensive treatment at baseline, osilodrostat therapy allowed for the reduction of the dose or the number of antihypertensive drugs in four of them. Mean (SD) QTcF did not significantly change during the observation: at baseline it was 394.17 (18.04) ms, and at week 156, 404 (17.01) ms.

Mean (SD) ALT and AST activity decreased during the whole study period: ALT at baseline was 29.33 (15.51) U/L, and at week 156, 15.33 (4.55) U/L; AST baseline was 23.33 (7.87) U/L, and at week 156, 18.33 (2.66) U/L. The fastest decrease in ALT and AST activity was observed during the first 36 weeks; however, the lowest values were achieved during the last follow-up.

Data are summarized in Table 3.

### 3.3. Treatment Tolerability and Adverse Events

Treatment with osilodrostat was generally well tolerated. During 156 weeks of observation, every patient experienced at least one adverse event (AE) suspected to be related to osilodrostat. Overall, all six patients experienced the AE of increased ACTH concentration. Five of the six patients (83.33%) presented with AI; there was no need to discontinue osilodrostat in any of these patients, and the dose was temporarily adjusted. A transient increase in 11-deoxycorticosterone (DOC) was observed in all patients, mainly after initiating osilodrostat treatment or during dose escalation. However, only two of the patients presented with the clinical effect of DOC accumulation (hypertension and hypokalemia in patient two and hypokalemia in patient four). All four female patients had a periodic increase in testosterone level, but only one presented with induced mild hirsutism, which resolved with time on continued treatment. Arthralgia, myalgia, or fatigue occurred in three of the six patients (50%), nausea, decreased appetite or hypokalemia in two patients (33.33%), and headache or hypertension in one patient (16.67%).

### 3.4. Description of Specific Cases

Patient #1

The first patient we would like to detail is a 28-year-old woman with persistent CD, complicated by obesity, decreased bone mineral density (BMD), impaired glucose tolerance, and arterial hypertension. Diagnostics found an onset in adolescence (at 16 years of age) when a non-intentional weight gain of 10 kg in six months with central and facial redistribution of body fat occurred, accompanied by acne and purple striae on the thighs and lateral surfaces of the trunk. At that time, the body weight centile chart corresponded to the 97th percentile, while height was between the 50th and 75th percentiles. The hormonal evaluation confirmed ACTH-dependent hypercortisolemia and an MRI visualized a 3 × 5 mm lesion of weak contrast enhancement in the posterior part of the pituitary gland. The patient was diagnosed with CD and, in June 2011, underwent a successful, uncomplicated TSS with biochemical and clinical remission, requiring six months of hydrocortisone replacement. Pathology examination identified densely granulated corticotroph adenoma with Ki-67 > 3%. About two years after the neurosurgery, the patient noticed weight gain, discoloration of the skin all over her body and reported secondary amenorrhea. The hormonal evaluation revealed CD recurrence. A follow-up MRI did not visualize a lesion compatible with a pituitary adenoma. In September 2013, the patient underwent an unsuccessful surgical exploration of the sella turcica without biochemical remission in a postoperative assessment.

Due to the persistent uncontrolled hypercortisolemia, the patient required medical therapy. In 2015–2016, the patient was treated with long-acting release (LAR) pasireotide. However, therapy had to be discontinued because of the induced hyperglycemia. In the following years, 2016 and 2017, the patient was treated with levoketoconazole, but the therapy was discontinued due to asymptomatic QT interval prolongation on ECG. In 2017–2018, the patient was switched to ketoconazole in combination with cabergoline; however, she reported continued weight gain, increased stretch marks on the abdominal skin, and persistent amenorrhea. Ketoconazole therapy had to be stopped due to its unavailability in Poland, while cabergoline was discontinued because of low effectiveness. Considering comorbid arterial hypertension and the patient’s reported deterioration of visual acuity, she was assessed ophthalmologically and diagnosed with grade I vascular retinopathy.

Due to the lack of clinical and biochemical control of the primary disease and gradually increasing complications of hypercortisolemia, the patient was offered to participate in the phase III LINC-4 trial. At the time of enrollment (September 2018), the hormonal tests were as follows: morning TSC—290.0 nmol/L (WRR), morning ACTH—2.9. pmol/L (WRR), mUFC—322.8 nmol/day (2.3× ULN), mLNSC—13.65 nmol/L (5.5× ULN). According to randomization, the patient was allocated to a placebo group and started an osilodrostat of 2 mg BID in the 12th week of the study. On the 14th week, laboratory assessment revealed TSC 546 nmol/L (WRR), mLNSC 12.15 nmol/L (5× ULN), and mUFC 433.4 nmol/d (3× ULN); therefore, the dose was increased to 5 mg BID, then in week 17 and 23 to 10 and 20 mg BID, respectively, which was maintained through the end of the week 47 of the study. The first significant decrease in hypercortisolemia severity was obtained at week 23: TSC 257 nmol/L (WRR), mLNSC 5.1 nmol/L (2× ULN), mUFC 174.85 nmol/d (1.3× ULN), and treatment was continued at a daily dose of 40 mg. The normalization of mUFC (113.55 nmol/d; WRR) and mLNSC (1.95 nmol/L; WRR) was achieved on the 32nd and 60th weeks of the study, respectively. In subsequent weeks, according to the results of hormonal tests and the patient’s clinical assessment, doses were modified appropriately and oscillated between 40 and 60 mg. After 156 weeks of treatment, a significant decrease in mUFC to 17.2 nmol/d (WRR) was achieved.

The patient tolerated the treatment well. There were no signs of adrenal insufficiency, except at week 132, when in the hormonal evaluation a significant reduction in morning TSC was found (75 nmol/L, <lower limit of normal). The dose of osilodrostat was reduced to 20 mg BID and continued after that until week 156 of the study. The patient episodically reported a mild degree of fatigue and mild intermittent headaches. The ECG demonstrated a mild prolongation of the QTc interval, which did not significantly progress in the following weeks of the study with treatment at a maximum daily dose of 60 mg. After the introduction of osilodrostat, testosterone levels increased significantly during dose escalation (maximum at week 48: 12.21 nmol/L—7.8x ULN, 3.67-fold increase compared to baseline); however, the patient did not present clinically significant hyperandrogenism (Ferriman and Gallwey score rated between 0 and 2 during the whole 156 weeks’ observation).

During the treatment with osilodrostat, the patient’s waist circumference decreased, and moderate weight reduction occurred, with the complete normalization of fat redistribution in the neck and supraclavicular regions and resolution of other cushingoid features—atrophy of proximal muscle groups and central obesity.

Patient #2

The second example of a patient whose clinical case we would like to detail is a 33-year-old male whose diagnostics started in September 2014 due to rapid unintentional weight gain with central fat redistribution, facial plethora, and purple striae. CS was suspected, and the baseline hormonal evaluation confirmed ACTH-dependent hypercortisolemia: positive corticotropin-releasing hormone (CRH) stimulation test, anterior pituitary lesion measuring 3 × 2.5 × 2 mm visualized in MRI, and positive bilateral inferior petrosal sinus sampling combined with CRH stimulation were compatible with CD. Other hormonal findings included thyrotropic and gonadotropic insufficiency. In November 2014, the patient underwent successful, uncomplicated TSS and required transient hydrocortisone substitution. In regular subsequent follow-ups, there was no recurrence of hypercortisolemia. Similarly, no tumor regrowth was found on the control MRI.

In February 2018, the patient reported a weight gain of 20 kg over the past few months and noticeably high blood pressure in in-home measurements. The hormonal evaluation revealed ACTH-dependent hypercortisolemia and a positive result of the combined LDDST and desmopressin test (CDDT) indicated the recurrence of CD. An MRI revealed a 2 mm focus of weak contrast enhancement on the left side of the anterior pituitary gland. In August 2018, the patient underwent a second TSS; however, the suppression of cortisol secretion was not achieved, compatible with persistent CD. Over the subsequent three months, the clinical features of hypercortisolemia dramatically intensified, including weight gain of another 10 kg, easy skin wounding, pustular skin lesions, and new purple striae. Based on an oral glucose tolerance test (OGTT), the patient was diagnosed with impaired FPG, and treatment with metformin was introduced. Hypotensive treatment with ramipril was also continued. Due to the refractory CD, the patient was offered to participate in the phase III LINC-4 trial. At the time of enrollment (December 2018), the hormonal tests were as follows: morning TSC—500 nmol/L (WRR), morning ACTH—5.3 pmol/L (WRR), mUFC—3316.4 nmol/day (24× ULN), mLNSC—20.85 nmol/L (8.35× ULN). According to randomization, the patient was allocated to the osilodrostat group. Based on the study protocol, he initially received osilodrostat at a dose of 2 mg BID, which then was adjusted every 4 weeks with an escalation sequence of 5–10 mg BID until week 12 when the dose was restarted. Throughout the open-label phase of the study, the dose of osilodrostat was systematically increased from the initial 2 mg to 30 mg BID according to the results of hormonal evaluations and the investigators’ clinical assessments. Several periods require particular comments during treatment. Significant improvement in hypercortisolemia severity was achieved after just 12 weeks of the treatment. In the hormonal evaluation performed on week 14, mUFC reduced nearly 5.5-fold from the baseline (611.95 nmol/L; 4× ULN). A normalization of mUFC and mLNSC was achieved in weeks 20 and 26 of the study, respectively, with a dose of osilodrostat of 20 mg BID. However, at week 26, rapid reduction in hypercortisolemia severity was found in hormonal evaluation (mUFC 52.45 nmol/day, LNSC 1.55 nmol/L, TSC 157 nmol/L), and as a result the dose of osilodrostat was reduced to 15 mg BID, although that time patient did not report any features of adrenal insufficiency.

Noteworthy is the favorable metabolic effect as early as the 14th week—a decrease of 10 kg in body weight and an improvement in metabolism parameters—producing a reduction in TC and FPG. However, treatment with metformin was also continued at that time. During the first 8 weeks of the study, a mild transient increase in ALT (67 U/L) activity was observed, which did not worsen during the subsequent increase in the daily dose, even to a maximum of 60 mg during the open-label phase of the trial.

The patient tolerated the treatment well. At week 56, the patient reported headache, persistent fatigue, and muscle and joint pain. Considering the reported symptoms of AI, the dose of osilodrostat was reduced, but then based on the increased mUFC from two subsequent visits and the absence of symptoms of AI, the dose was increased again to 25 mg BID on week 64, which the patient tolerated well. A relevant increase in blood pressure was observed in week 108 of the study along with mild hypokalemia. Laboratory tests at that time revealed significantly elevated DOC concentration: 5447 pmol/L (12× ULN, RR: <454 pmol/L). The dose of ramipril taken was increased, and spironolactone was included. However, during following weeks the hypertension control worsened as the DOC concentration increased even more (during osilodrostat up-titration to maximum dose of 60 mg)—at week 132 it was 14 253 pmol/L (31.4× ULN)—meaning further antihypertensive and mineralocorticoid receptor blockade treatment intensification was required, which resulted in satisfactory BP control at week 156. DOC concentration at that the end of the observation declined to 3693 pmol/L (8.15× ULN).

Treatment with osilodrostat resulted in spectacular improvements both biochemically and clinically. After 156 weeks of treatment, a pleasing decrease in mUFC to 110.6 nmol/d (WRR) was demonstrated. Therapy with osilodrostat also resulted in excellent enhancement in clinical features of hypercortisolemia—fat redistribution and muscle atrophy disappeared, and the severity of striae decreased.

Patient #3

The next patient presented is a 23-year-old woman with hypercortisolemia symptoms appearing during pregnancy, when she experienced significant weight gain (approximately 20 kg) with purple striae and developed gestational hypertension. Postpartum, secondary amenorrhea appeared, striae intensified, and the patient presented with hirsutism, acne, and a tendency to easy bruising. Furthermore, there was a central body fat redistribution and moderate muscle atrophy with weakness. The patient was referred to the department of endocrinology, and in December 2013, she was diagnosed with ACTH-dependent hypercortisolemia. Additional comorbidities included diabetes mellitus and osteoporosis. An MRI revealed a hypointense focal pituitary lesion measuring 6x4 mm. Pending surgery, bridging therapy with ketoconazole was implemented, and TSS was successfully performed in January 2014.

After four years of remission, from January 2018, the patient showed signs of CD recurrence, e.g., weight gain with central fat distribution, generalized weakness, muscle fatigue, hirsutism, acne, purple striae, tachycardia, and hypertension. The hormonal evaluation confirmed ACTH-dependent hypercortisolemia, and positive high-dose DST and CDDT were compatible with CD. An MRI revealed a small (1.5 mm) hypointense lesion on the left side of the anterior pituitary. In September 2018, the patient underwent a second TSS; however, the postoperative hormonal evaluation showed persistent hypercortisolemia. Due to the refractory CD, the patient was offered to participate in the phase III LINC-4 trial. At the time of enrollment (November 2018), the hormonal tests were as follows: morning TSC—328.0 nmol/L (WRR), ACTH—10.0 pmol/L (WRR), mUFC—380.03 nmol/24 h (2.75× ULN), mLNSC—7.35 nmol/L (2.94× ULN). According to randomization, the patient was allocated to the placebo group and started on osilodrostat of 2 mg BID in the 12th week of the study. Initially, we observed a significant improvement in clinical symptoms and biochemical control of the disease; mUFC and mLNSC had normalized only after 5 and 8 weeks, respectively. The response to the treatment was highly satisfactory, and the patient periodically required a dose reduction to 1 mg daily. At week 36 of the study, her body weight had decreased by 6 kg. At that time, mUFC and mLNSC were 34.97 nmol/24 h and 0.80 nmol/L, respectively. However, a pituitary MRI performed at week 48 showed tumor size progression (9 × 5 × 8 mm). ACTH at that time was 24 pmol/L (2.16× ULN).

From week 60, the patient noticed a gradual increase in body weight, which was assumed to be associated with the course of the primary disease, evidenced by an increase in mUFC (421,60 nmol/24 h, 3.06× ULN); therefore, the osilodrostat dose was increased. Unfortunately, conducting unscheduled visits with more detailed diagnostics was much more difficult in Poland at that time due to the ongoing COVID-19 pandemic. Due to epidemiological reasons, the visit in week 72 could only be conducted remotely. In the 84th week of the study, the physical examination showed weight gain (67 kg, BMI: 26.8 kg/m^2^) with central fat distribution, dorsal and supraclavicular fat pads, new abdominal purple striae, generalized hyperpigmentation, and an increase in BP. There was a sudden increase in mUFC (2878.75 nmol/24 h, 20.86× ULN), mLNSC (33.25 nmol/L, 13.3× ULN), and morning TSC (1046 nmol/L, 1.85× ULN). ACTH (182.7 pmol/L, 16.46× ULN) had significantly increased—by approximately 18.3-fold compared to baseline. Pituitary MRI revealed a further enlargement of the tumor (12 × 8 × 14 mm) extending to the left sinus cavernous, adjacent to the left internal carotid artery, without chiasmatic compression. The patient reported no headaches or visual disturbances.

Due to the corticotroph tumor progression (CTP), the patient was discussed during a multidisciplinary pituitary tumor board and qualified for the next operation. However, it was decided to continue osilodrostat treatment at an increased dose (5 mg BID) to control hypercortisolemia. The patient underwent a third TSS in October 2020 (week 100), however, it was possible to only partially remove the tumor. Morning TSC on the first postoperative day (511.18 nmol/L, WRR) showed persistent CD, and it was decided to continue osilodrostat. Pathology examination identified corticotroph adenoma with a high proliferation index (Ki-67 > 10%). A controlled pituitary MRI, performed in week 120, showed even further tumor progression: 14.5 × 8.5 × 15.5 mm. In June 2021 (week 136), the patient underwent a single-session *Gamma Knife* stereotactic radiosurgery for the residual tumor at a dose of 20 Gy. From the onset of CTP, osilodrostat was maintained at 5 mg BID, which allowed the disease to be controlled (mUFC < ULN) from week 120. At week 156 (at the end of this study observation), the patient reported fatigue and decreased appetite; low mUFC (33.05 nmol/24 h, WRR) indicated AI, and the osilodrostat was reduced to 2 mg BID. The MRI showed a slight regression of tumor remnant (11.5 × 7 × 13 mm). The patient remains eucortisolemic to date on a small dose of osilodrostat.

## 4. Discussion

CD remains a major diagnostic and therapeutic challenge. Uncontrolled, chronic hypercortisolemia is associated with significant morbidity, including metabolic complications comprising glucose metabolism impairment and dyslipidemia, arterial hypertension, hypercoagulability, increased risk of infections, and psychoneurological complications, which overall results in impaired quality of life, increased mortality, and reduced life expectancy compared to that of the general population. The goal of the treatment is to reverse signs and symptoms of hypercortisolemia by achieving cortisol normalization, improving quality of life, alleviating the burden of morbidity, and reducing mortality.

TSS is a treatment of choice in CD with a promising remission rate (70–90%); however, long-term recurrence rates reach up to 35% [10,15]. Pharmacotherapy has had a secondary role in managing patients with CD, but thanks to the development of several novel drugs, it has become more relevant. The spectrum of available medications in CD includes adrenal steroidogenesis inhibitors, glucocorticoid receptor antagonists, and pituitary-directed drugs; however, they exhibit specific limitations [10,12,13,14]. In light of the unmet medication need for the optimal medical treatment of patients with CD, osilodrostat (a potent inhibitor of 11-β-hydroxylase synthase enzyme) has emerged. In clinical trials, osilodrostat provided rapid onset and long-term control of cortisol production, sustained control of biochemical parameters, and improved clinical signs and physical manifestations of hypercortisolemia and patient-reported outcomes.

Smaller studies depicting the cases of patients in real-life clinical settings can be valuable supplements to large clinical trials. This report documents the course of osilodrostat therapy in a group of six Polish patients previously taking part in the LINC4 study and indicates its effectiveness as a long-term treatment option for patients with CD. The results of our observational study are consistent with those obtained during the clinical trials. Osilodrostat allowed for achieving biochemical control, resolution of clinical symptoms of hypercortisolemia, and improvement in the quality of life of all patients in the study. By week 36, mUFC < ULN was achieved in all patients at a median osilodrostat dose of 4 mg BID. At the end of the observation, mUFC normalization was sustained in all patients at a median osilodrostat dose of 3 mg/day. At week 96 (LOV for mLNSC), 4/6 patients had mLNSC < ULN. In all patients, the highest mUFC and mLNSC level reduction was observed in the first 36 weeks of the treatment (*p* = 0.031). Since mUFC and mLNSC seem to be equal in assessing the efficacy of pharmacological treatment of CD and the best prognosis is associated with normalization of both parameters [20], complete biochemical control (defined as both parameters < ULN) at week 96 was achieved in 4/6 patients. However, all patients presented with normalization of mUFC and mLNSC at least temporarily, and after week 96, assessing mLNSC was no longer possible. There was no apparent relationship between baseline mUFC or mLNSC and the osilodrostat dose required for mUFC or mLNSC normalization, and the dose range to maintain disease control was 1 mg every three days to 30 mg BID.

All patients demonstrated significant improvement from baseline in most metabolic parameters, including weight, BMI, waist circumference, TC, TG, FPG and HbA1c, and this was most evident at week 36 and then sustained throughout the study period. The osilodrostat allowed for discontinuation of antilipidemic drugs in one of the three and hypoglycemic treatment (metformin) in two of three (one diabetic and one with IGT) patients taking the drugs at baseline. Improvement in BP control was most evident early in osilodrostat treatment: the most significant decrease in mean SBP and mean DBP was achieved at weeks 72 and 36, respectively. However, from week 96, a slight increase in mean SBP and mean DBP was observed, and a maximum mean SBP and mean DBP during the study were observed at weeks 132 and 156, respectively. This change was probably because, in 4/5 patients under antihypertensive treatment at baseline, osilodrostat allowed for the dose or number of the drugs to be reduced. On the other hand, one hypertensive patient at baseline presented high BP between weeks 108 and 144 of the study due to the osilodrostat dose escalation and the DOC accumulation effect, requiring intensifying antihypertensive treatment (including implementation of mineralocorticosteroid blockage). However, mean SBP and BDP remained WRR during the whole study period. Of course, the analysis of BP, lipid parameters, FPG, and HbA1c in the presented patients has limited value, mainly since the patients were taking antihypertensive, antilipidemic, and/or hypoglycemic drugs before osilodrostat initiation. Clinically significant QTcF prolongation (≥450 ms) did not occur in any of the patients.

The three cases presented in detail confirm high osilodrostat effectiveness in CD management and highlight the specific clinical situations that physicians may have to face during osilodrostat therapy. In the first case, osilodrostat was the “last chance” medical treatment, considering the necessity to discontinue the pasireotide and levoketoconazole because of their side effects and unsuccessful combination therapy with ketoconazole and cabergoline with further hypercortisolemia progression. The patient experienced one of the expected osilodrostat AE because, after its initiation and dose titration, testosterone levels increased significantly. However, clinically the patient did not present hirsutism or other hyperandrogenic features. Increased testosterone concentration in females during osilodrostat treatment have been reported in 11.7–24.7% during clinical trials [18,21]. It results from the shift of adrenal steroidogenesis towards androgens after the inhibition of cortisol production [22]. In the presented patient, testosterone level then returned to baseline with time on continued osilodrostat after longer follow-up (week 144). That observation is consistent with data from extension phases of LINC studies, where increased testosterone levels in females resolved during long-term treatment [19,21,23]. Nevertheless, hirsutism in most CD females during osilodrostat treatment improves or remains stable from baseline [18,24].

The second of the presented patients illustrated a spectacular improvement in uncontrolled, refractory hypercortisolemia upon osilodrostat treatment. The osilodrostat allowed for a normalization of mUFC (24× ULN at basline) and mLNSC (8.35× ULN at baseline) after 20 and 26 weeks, respectively, which confirms its effectiveness, even in patients with a more severe course of the disease. As in the previously discussed case, this patient also experienced osilodrostat AE related with 11β-hydroxylase enzyme blockade—a significant adrenal hormone precursor increase (the concentration of DOC exceeded 31× ULN at one point) manifesting as hypertension and mild hypokalemia that occurred from week 108. The patient required further antihypertensive treatment intensification with implementation of spironolactone, which allowed for BP re-control at week 156, and oral potassium supplementation. Mineralocorticoid precursor increase-related AEs have been reported in patients during clinical trials, represented by hypertension (12.4–21.9%), hypokalemia (13.1%), and peripheral edema (15.3–16.4%) [17,18,21].

The third patient we presented, after an initial good response to osilodrostat treatment, developed a significant COP after 72 weeks, requiring a multimodal approach, including osilodrostat up-titration, surgery, and radiosurgery. Tumor size changes have been previously reported with osilodrostat. In the LINC-3 study, tumor volume decreased or increased significantly (by ≥20%) in 32.8% and 37.5% of CD osilodrostat-treated patients, respectively [17]. Fontaine-Sylvestre et al. described a specific LINC-3 trial case of CTP requiring surgical intervention after long-term treatment (over four years) with osilodrostat for persistent CD [25]. Antonini et al. presented a patient with significant CTP after six months of osilodrostat therapy that was eventually treated with radiosurgery [26]. CTP was also reported during treatment with ketoconazole [27,28] and mifepristone [29], which suggests that blocking cortisol production or action by medications may provoke the growth of the corticotroph adenoma. It is suggested that osilodrostat therapy should be stopped if CTP is observed [30]; however, no specific recommendations are available. In the described case, we decided to continue osilodrostat and increased the daily dose to stabilize the hypercortisolemia and the patient’s clinical condition, which allowed for safe subsequent treatment procedures (TSS, pituitary radiosurgery) and resulted in CD re-control. Controversy exists about whether CTP during osilodrostat treatment is caused by loss of negative cortisol feedback (due to a decrease in cortisol concentration) or whether it is driven by genetic mechanisms implicated in corticotroph tumor development, implying that it is a part of the natural history of CD. In vitro studies did not show that osilodrostat had an impact on ACTH production or cell growth [31]. However, the case we presented and previously mentioned by Fontaine-Sylvestre et al. [25] and Antonini et al. [26] implies that regular biochemical and radiological monitoring is necessary for patients treated with osilodrostat, even when they are well controlled on a stable osilodrostat dose. In addition, an increase in ACTH concentration was observed in each of the patients participating in the study. Median ACTH concentration significantly increased during the whole study period, and at 156 weeks’ follow-up (91.6 pmol/L, 8.25× ULN, range: 11.1–200 pmol/L), it was 9.49-fold higher (*p* = 0.031) compared to baseline (9.65 pmol/L, WRR, range: 2.9–28.60 pmol/L). Detailed analysis of tumor volume was not possible in the presented group; however, in 2/6 patients the tumor remained stable, in 2/6 patients there was a significant (>20%) increase in tumor volume, and in 2/6 patients there was a radiological disappearance of the initially visible tumor. There was no clear correlation between change in tumor volume and osilodrostat dose nor ACTH concentration. These data are consistent with those observed in clinical trials, indicating that the effect of osilodrostat on corticotropic tumor size is ambiguous and requires further study. However, data show that there is no correlation between change in tumor volume and change in ACTH concentration [19]. The evolution of pituitary tumor in every patient is summarized in Table 4.

The second most common AE reported in the 156 weeks of osilodrostat treatment was AI (5/6 patients). AI/hypocortisolism-related AEs have been reported in 14.6–54.0% of patients, mainly during osilodrostat titration [16,17,18,21,23]. However, the percentage of patients experiencing AI was increasing in longer follow-up and occurred even in patients on a stable dose of osilodrostat after prolonged treatment [23,32]. AI events in the analyzed group generally occurred during the titration phase and were not associated with a specific osilodrostat dose, which is consistent with previous observations [33]. In two patients, AI occurred after a longer time of observation (as presented earlier); however, in the third of the detailed cases, the late AI occurrence might not have been strictly related to osilodrostat but was the effect of concomitant neuro- and radiosurgery. None of the patients discontinued osilodrostat because of the hypocortisolism—dose adjustment and in some cases transient hydrocortisone implementation was sufficient. Ours and previously reported experience indicates that patients treated with osilodrostat should be thoroughly educated about its potential side effects (especially the risk of AI and related symptoms) and supplied with “in case” hydrocortisone tablets, as hypocortisolemia can occur at any time during osilodrostat treatment [30,34,35]. In the presented patients, it was sufficient to administer hydrocortisone for a few days at a typical replacement dose and reduce the osilodrostat dose. The osilodrostat dose was then readjusted to achieve eucortisolemia. However, given osilodrostat’s high effectiveness in hypercortisolemia control and, therefore, the relatively high risk of AI, a “block and replace” approach should be considered in specific cases. However, AI is not specific to osilodrostat AE, and the possibility of its occurrence in patients treated with oral steroidogenesis inhibitors in general (5.3–18.5% with ketoconazole, 3.2–9.5% with levoketoconazole, 12% with metyrapone) should be expected [13,14]. The higher incidence of AI during osilodrostat treatment is probably due to its particular potency and 11-beta-hydroxylase selectivity compared to other steroidogenesis inhibitors. It is worth mentioning that escaping after the initial response may occur in up to 22.7% of patients treated with ketoconazole and 18.7% with metyrapone [13,14]. Furthermore, the safety profile of osilodrostat appears to be favorable. The main limitations of using ketoconazole and levoketoconazole are hepatotoxicity and elevations in liver enzymes, which are reported in 2.6–18.4% and 11.7–44.6% of patients, respectively [13,14]. In the analyzed group, only one patient had a mild, transient increase in liver enzyme activity >ULN during the study. Mean ALT and AST activity even decreased compared to baseline during the study period in the analyzed group. No liver-related AEs of osilodrostat were reported in previous studies. [16,17,18,21,23]. Adrenal androgens and mineralocorticoid precursor increase-related AEs are the main limitations of metyrapone use; however, they may occur in 42.3–58.4% of patients on osilodrostat, as previously mentioned [14,21]. Nevertheless, Bonnet-Serrano et al. showed a smaller increase in 11-deoxycortisol and androgen concentrations in patients with osilodrostat compared with metyrapone therapy [36]. Additionally, mean levels of adrenal hormone precursors and androgens decreased during long-term osilodrostat treatment [23]. Osilodrostat also has a longer half-life compared with metyrapone and ketoconazole, which allows for BID administration and may increase patient compliance [12,13,14]. Moreover, ketoconazole and levoketoconazole are unavailable in Poland at the moment. In specific scenarios (especially of severe hypercortisolemia), osilodrostat in combination with other anticortisolic agents can be considered; however, data on such an approach are limited [37,38].

## 5. Conclusions

Despite progress in the treatment of CD, it remains a real challenge for endocrinologists and neurosurgeons, especially in the case of recurrent disease and when patients cannot feasibly undergo surgery. In light of the unmet need for the optimal treatment of patients with CD, osilodrostat emerged as a novel steroidogenesis inhibitor. The analyzed group of patients shows the long-term effectiveness and safety of the treatment with osilodrostat in real-life clinical practice, as treatment in all patients was reimbursed by the Health National Fund after completing the clinical trial. Our study reinforced previous reports demonstrating that osilodrostat is an effective and well-tolerated treatment option for patients with CD and provides sustained hypercortisolemia control. Treatment with osilodrostat should be individualized for each patient, and the clinical evidence gathered through this study might assist in optimizing treatment decisions by physicians.

## Figures and Tables

**Figure 1 biomedicines-11-03227-f001:**
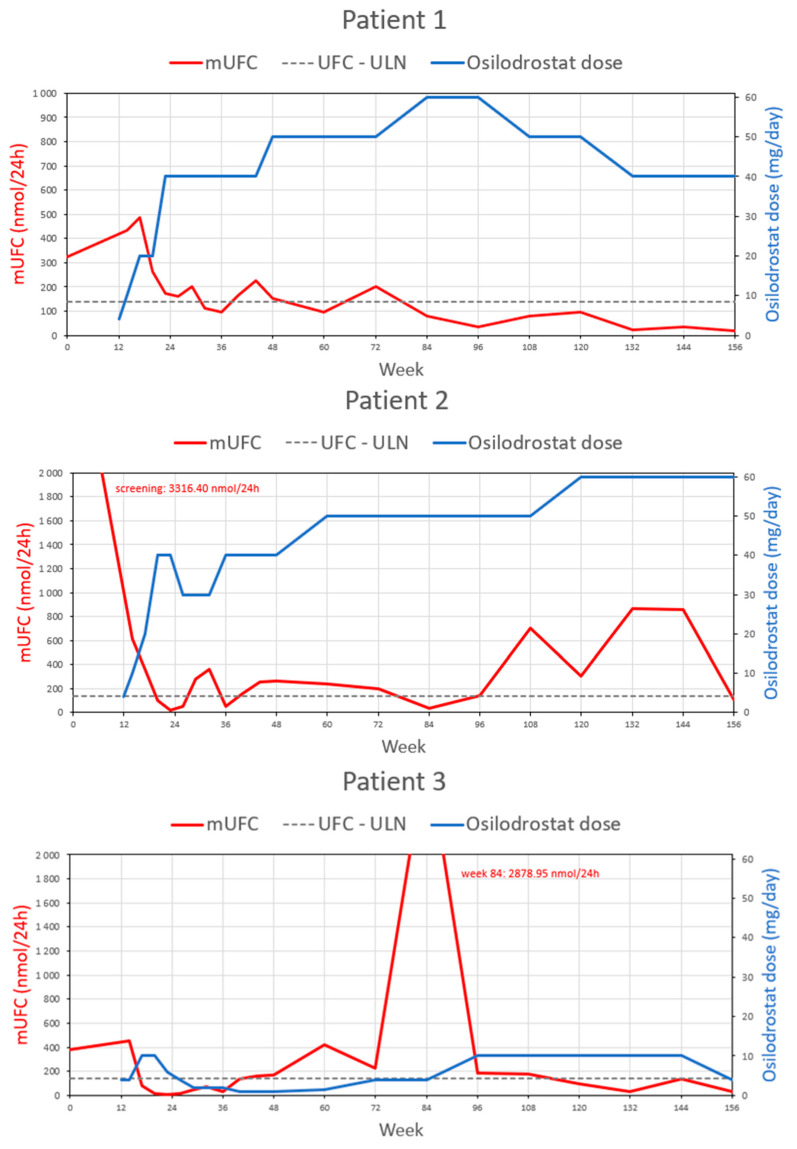
mUFC evolution according to osilodrostat dose during 156 weeks’ observation for every patient.

**Figure 2 biomedicines-11-03227-f002:**
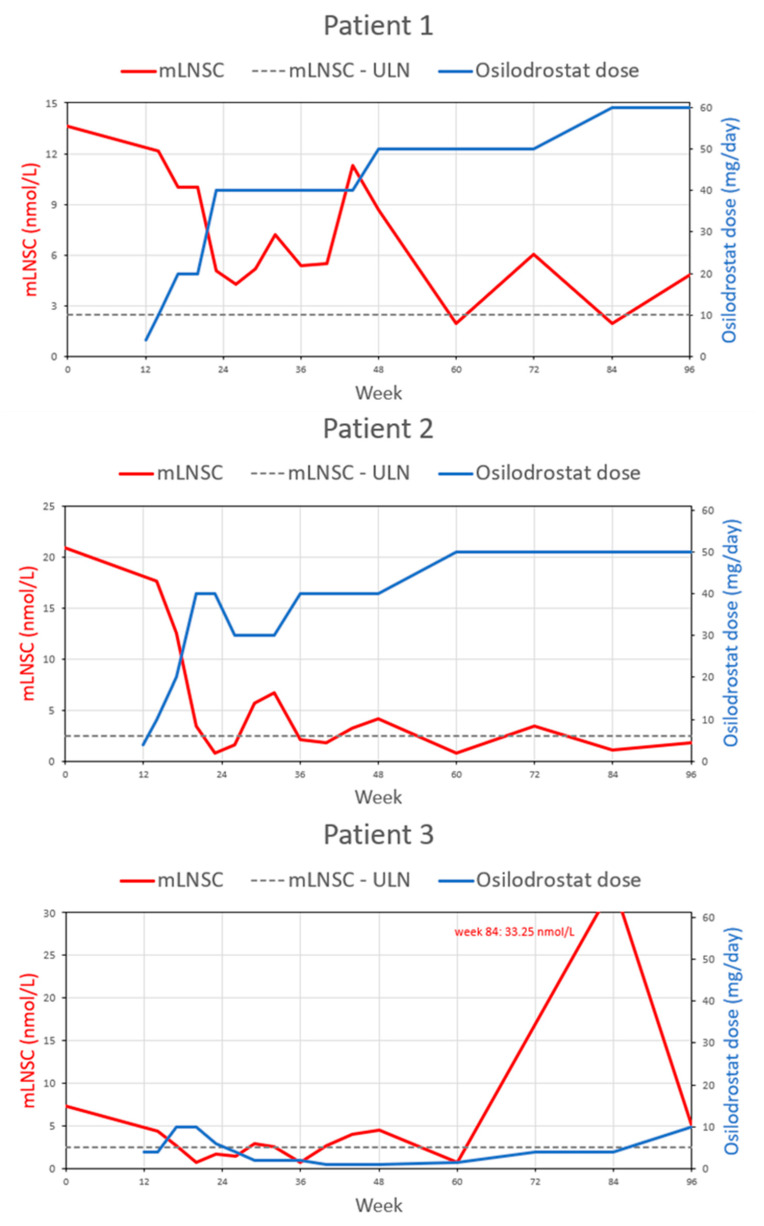
mLNSC evolution according to osilodrostat dose during the 96 weeks’ observation for every patient.

**Figure 3 biomedicines-11-03227-f003:**
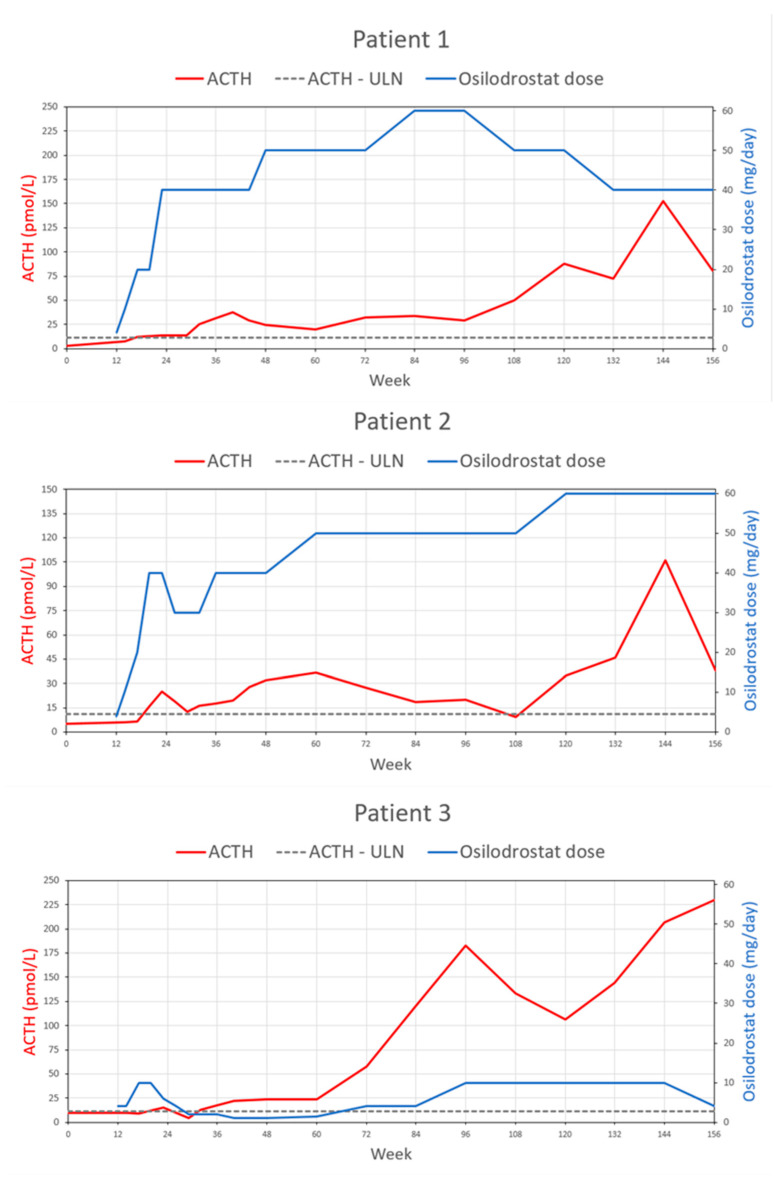
ACTH evolution according to osilodrostat dose during 156 weeks’ observation for every patient.

**Figure 4 biomedicines-11-03227-f004:**
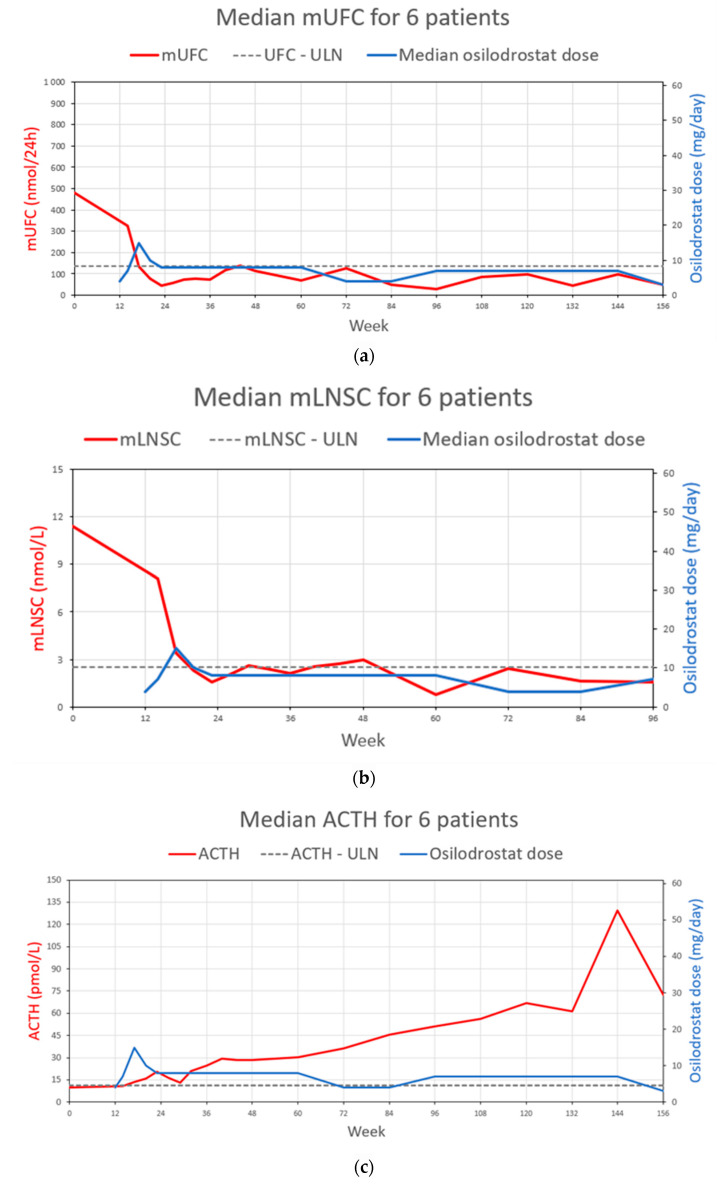
Median mUFC (**a**), median mLNSC (**b**), and median ACTH (**c**) evolution according to median osilodrostat dose during the study observation of all patients.

**Table 1 biomedicines-11-03227-t001:** Baseline patient characteristics.

Characteristics	All Patients, *n* = 6
Age, years	
Median (range)	32.5 (24–62)
Mean (SD)	35.7 (12.8)
Sex, *n*	
Female	4
Male	2
Previous treatment, *n* (%)	
Pituitary surgery	6
Medical therapy	5
Comorbidities, *n* (%)	
Hypertension	5
Abnormal weight	
Overweight	4
Obesity	1
Impaired glucose metabolism	
IFG	1
IGT	2
DM	2
Dyslipidemia	5
Decreased BMD	5
Physical manifestation of hypercortisolemia, n	
Central obesity	6
Dorsal fat pad	5
Supraclavicular fat pads	6
Facial rubor	3
Proximal muscle atrophy	4
Striae	6
Ecchymoses	2
Hirsutism (in females, *n* = 4)	2
mUFC, nmol/24h	
Median (range)	480.0 (220.5–3316.4)
Mean (SD)	900.3 (1839)
mLNSC, nmol/L	
Median (range)	11.4 (3.60–28.85)
Mean (SD)	11.4 (5.35)

IFG—impaired fasting glucose; IGT—impaired glucose tolerance; DM—diabetes mellitus; BMD—bone mineral density; mUFC—mean urinary free cortisol; mLNSC—mean late-night salivary cortisol.

**Table 2 biomedicines-11-03227-t002:** Median mUFC, mLNSC, TSC, and ACTH concentrations and their changes from baseline at each checkpoint during observation.

Follow-Up	Median (Range)	Δ% from Baseline	*p*-Value
mUFC (nmol/24 h)			
Baseline	480.02 (220.53–3 316.40)	-	-
Week 36	75.15 (34.97–94.60)	−87.60	0.031
Week 72	127.83 (17.80–223.70)	−90.73	0.031
Week 96	30.26 (8.90–182.40)	−95.77	0.031
Week 132	44.48 (22.40–868.30)	−89.32	0.031
Week 156	50.83 (17.20–110.60)	−92.99	0.031
mLNSC (nmol/L)			
Baseline	11.43 (3.60–20.85)	-	-
Week 36	2.11 (0.80–5.35)	−77.05	0.031
Week 72	2.43 (0.80–6.05)	−83.45	0.031
Week 96	1.58 (0.80–5.10)	−82.75	0.031
TSC (nmol/L)			
Baseline	448.5 (290.0–673.0)	-	-
Week 36	268.0 (196.0–367.0)	−41.75	0.063
Week 72	263.0 (102.0–511.0)	−49.73	0.156
Week 96	331.0 (86.0–820.0)	−56.42	0.438
Week 132	307.5 (75.0–392.0)	−42.26	0.031
Week 156	281.5 (116.0–420.0)	−33.64	0.031
ACTH (pmol/L)			
Baseline	9.65 (2.90–28.60)	-	-
Week 36	29.45 (17.10–46.20)	+161.23	0.031
Week 72	37.00 (13.30–245.80)	+362.09	0.031
Week 96	56.30 (9.50–242.00)	+677.09	0.031
Week 132	129.55 (14.20–264.60)	+1933.44	0.031
Week 156	91.60 (11.10–200.00)	+651.29	0.031

mUFC—mean urinary free cortisol; mLNSC—mean late night salivary cortisol; TSC—total serum cortisol; ACTH—adrenocorticotropic hormone.

**Table 3 biomedicines-11-03227-t003:** Metabolic, cardiovascular, and liver parameters at baseline and their changes from baseline at checkpoints during observation.

Parameter (Mean, SD)	Baseline	Week 36	Δ%	Week 72	Δ%	Week 96	Δ%	Week 132	Δ%	Week 156	Δ%
Weight (kg)	69.88	64.72	−6.49	64.68	−6.75	64.20	−7.48	64.92	−6.51	63.75	−8.66
	(13.52)	(8.38)	(7.47)	(9.40)	(6.79)	(10.04)	(8.52)	(9.68)	(5.80)	(11.79)	(4.52)
BMI (kg/m^2^)	26.74	24.87	−6.49	24.83	−6.75	24.64	−7.48	24.90	−6.51	24.41	−8.67
	(3.40)	(2.30)	(7.47)	(2.55)	(6.79)	(2.83)	(8.52)	(2.46)	(5.80)	(3.13)	(4.52)
Waist circumference	93.67	89.92	−3.99	88.67	−5.49	89.92	−7.34	86.40	−8.79	84.83	−9.50
(cm)	(3.93)	(4.03)	(2.32)	(9.27)	(6.88)	(9.78)	(7.38)	(9.71)	(7.96)	(9.06)	(7.79)
SBP (mmHg)	125.00	115.67	−7.25	113.00	−9.29	119.17	−4.30	130.33	+4.56	121.17	−2.76
	(8.25)	(8.60)	(7.61)	(12.18)	(11.46)	(16.23)	(14.63)	(16.75)	(14.51)	(3.71)	(6.18)
DBP (mmHg)	79.17	76.25	−3.11	76.50	−2.75	80.17	+2.01	83.00	+4.93	83.33	+1.99
	(7.20)	(7.24)	(12.57)	(6.72)	(12.28)	(11.44)	(17.20)	(14.14)	(15.59)	(7.09)	(11.07)
QTcF (ms)	394.17	407.83	+3.62	412.41	+4.73	411.08	+4.46	410.00	+4.05	404.17	+2.65
	(18.04)	(4.96)	(4.13)	(27.30)	(7.07)	(21.10)	(7.12)	(21.77)	(4.18)	(17.01)	(4.96)
FPG (mmol/L)	5.12	4.53	−8.43	4.47	−11.22	4.33	−12.87	-	-	-	-
	(1.23)	(0.48)	(16.49)	(0.67)	(7.70)	(0.43)	(13.09)				
HbAlc (%)	5.55	5.35	−3.01	5.40	−2.19	5.33	−3.29	5.32	−3.84	5.35	−3.26
	(0.61)	(0.49)	(5.95)	(0.57)	(6.08))	(0.45)	(5.50)	(0.59)	(4.27)	(0.56)	(2.00)
TC (mmol/L)	5.58	4.38	−20.43	4.55	−14.75	4.02	−25.73	-	-	-	-
	(1.05)	(1.59)	(24.89)	(0.69)	(25.26)	(0.69)	(18.51)				
TG (mmol/L)	1.94	1.53	−25.16	1.31	−19.84	1.55	−1.87	-	-	-	-
	(1.21)	(1.70)	(35.60)	(0.72)	(42.82)	(0.86)	(52.12)				
ALT (U/L)	29.33	17.08	−23.59	22.83	−9.55	24.50	−14.61	18.50	−14.33	15.33	−30.27
	(15.51)	(4.41)	(48.42)	(12.48)	(83.47)	(15.83)	(92.09)	(6.29)	(50.99)	(4.55)	(40.88)
AST (U/L)	23.33	18.42	−11.75	20.17	−5.81	23.08	−7.20	19.83	−11.25	18.33	−15.06
	(7.87)	(2.80)	(36.55)	(4.79)	(34.70)	(10.39)	(54.01)	(7.25)	(26.28)	(2.66)	(25.58)

BMI—body mass index; SBP—systolic blood pressure; DBP—diastolic blood pressure; QTcF—corrected QT interval; FPG—fasting plasma glucose; HbA1c—glycated hemoglobin; TC—total cholesterol; TG—triglycerides; ALT—alanine transaminase, AST—aspartate transaminase.

**Table 4 biomedicines-11-03227-t004:** Evolution of pituitary tumor image on MRI during study period. The size of the tumor is presented in millimeters. In case of small pituitary tumors, only two or one (maximal) dimensions are given.

	Baseline	Week 26	Week 48	Week 72	Week 96	Week 144
Patient 1	4 × 3	No visible	No visible	No visible	4 × 3	5 × 3
Patient 2	10 × 13 × 10	10 × 13 × 10	10 × 13 × 10	10 × 13 × 10	14 × 16 × 14	14 × 16 × 14
Patient 3	1.5	3.5 × 5.5	7 × 5 × 8	12 × 8 × 14 *	14.5 × 8.5 × 15.5 **	11.5 × 7 × 13
Patient 4	2.5 × 4	3 × 4	2.5 × 4	2.5 × 4	no visible	no visible
Patient 5	4.5 × 4	4 × 3	3.5 × 2.5	3.5 × 2.5	3.5 × 2.5	3.5 × 2.5
Patient 6	3 × 2.5 × 2	2	2	2	2	no visible

* Performed at week 84. ** Performed at week 120.

## Data Availability

The data presented in this study are available on request from the corresponding author. The data are not publicly available due to the ethical reasons.

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
