# Peer review of "Cushing’s Disease: Long-Term Effectiveness and Safety of Osilodrostat in a Polish Group of Patients with Persistent Hypercortisolemia in the Experience of a Single Center"

_biomedicines, 2023, doi:10.3390/biomedicines11123227_

Round 1

Reviewer 1 Report

Comments and Suggestions for Authors

Comments
-The study confirms previous studies, but an important limitation is that a detailed analysis of tumor volume could not be assessed. Such an analysis should have been assessed in patients previously successfully operated on and now in recurrence
-Blockade of 11-hydroxylase results in increased DOC and thus pseudohyperaldosteronism. In Cushing's syndrome, pseudohyperaldosteronism is related to the saturation of 11HSD2 by cortisol, which becomes a mineralcorticoid, whereas therapy with osilodrtostat should create a similar situation this time related to the binding of DOC. to mineral0corticoid receptors. Report whether DOIC was found to be increased in only some or all patients.
-Explain whether the patients in Table 1 were without therapy with other drugs used for CD or for hypertension or for dysplipidemia
-Include UFC in the figures as well to see whether increases in LNFC were associated with increases in UFC
-ACTH increased in all patients, and thus the effect of therapy on possible Cushing's recurrence was not yet evident in many cases perhaps due to previous TTS, but certainly would be expected. It would be important to combine MRI of the pituitary gland.
-A promising drug is spironolactone, which also has multiple effects in Cushing's syndrome and particularly during therapy with 11-hydroxylase blockers. Spironolactone has an antialdosteronic action, reduces aldosterone synthesis in the adrenals, has a partial blocking action of 11-hydroxylase. An important factor to be reported is the type of therapy for hypertension, which should take into account the state of pseudohyperaldosteronism and thus more than ACE inhibitors should be aimed at antialdosteronic action
-Therapy for dyslipidemia, diabetes, and hypertension clearly interferes with assessment of baseline results reported in the tables
In conclusion, the study confirms the previous ones, and the factors reported in the comments would make it more complete

Author Response

"The study confirms previous studies, but an important limitation is that a detailed analysis of tumor volume could not be assessed. Such an analysis should have been assessed in patients previously successfully operated on and now in recurrence”

Unfortunately, in our opinion, a detailed analysis of the change in tumor size is difficult to perform. In some patients, the tumor was invisible at certain stages of observation. Hence, conclusions about a change in tumor size/volume for a group of patients are limited. However, we decided to present the change in tumor size on MRI for each patient separately in an additional table (Table 4.)

“Explain whether the patients in Table 1 were without therapy with other drugs used for CD or for hypertension or for dysplipidemia”

The results in table 1 refer to the parameters of patients without specific treatment for hypercortisolemia, but the patients were taking medications for comorbidities (hypertension, dyslipidemia, impaired glucose metabolism, decreased bone mineral density), at that time.

“Blockade of 11-hydroxylase results in increased DOC and thus pseudohyperaldosteronism. In Cushing's syndrome, pseudohyperaldosteronism is related to the saturation of 11HSD2 by cortisol, which becomes a mineralocorticoid, whereas therapy with osilodrostat should create a similar situation this time related to the binding of DOC to mineralocorticoid receptors. Report whether DOC was found to be increased in only some or all patients.”

Thank you for that comment. DOC levels were transiently increased in all six patients, mainly after initiating osilodrostat treatment or during dose escalation. However, only 2 of the patients presented with the clinical effect of DOC accumulation (hypertension and hypokalemia in patient two and hypokalemia in patient four). We included that information in the text. Additionally, we added information in the text regarding the influence of DOC and other precursors on the total serum cortisol concentration.

„Include UFC in the figures as well to see whether increases in LNFC were associated with increases in UFC”

Thank you for this comment; we initially planned to present mUFC and mLNSC in one graph, but follow-up for mLNSC was only available until week 96 due to technical reasons. We have, therefore, decided to present mUFC and mLNSC in separate graphs depending on the dose of osilodrostat.

„ACTH increased in all patients, and thus the effect of therapy on possible Cushing's recurrence was not yet evident in many cases perhaps due to previous TTS, but certainly would be expected. It would be important to combine MRI of the pituitary gland.”

An increase in ACTH concentration was observed in all patients. Still, we found no clear relationship between the degree of ACTH increase and the progression of the size of the corticotroph tumor in the analyzed group. For example, the highest ACTH concentration during observation was found in a patient in whom the initially visible focal lesion of the pituitary gland disappeared during observation (patient 6). However, we decided to present the change in ACTH concentrations over time for each patient and the median ACTH for all patients in separate graphs. In our opinion, the analysis of these graphs together with the data from the table regarding changes in the image of the pituitary gland in MRI will be sufficient for the reader.

“A promising drug is spironolactone, which also has multiple effects in Cushing's syndrome and particularly during therapy with 11-hydroxylase blockers. Spironolactone has an antialdosteronic action, reduces aldosterone synthesis in the adrenals, has a partial blocking action of 11-hydroxylase. An important factor to be reported is the type of therapy for hypertension, which should take into account the state of pseudohyperaldosteronism and thus more than ACE inhibitors should be aimed at antialdosteronic action”

Thank you for that comment. Indeed, we fully agree with this, and that’s why, in patient 2, spironolactone was included during the time of DOC-related hypertension.

„Therapy for dyslipidemia, diabetes, and hypertension clearly interferes with assessment of baseline results reported in the tables”

Of course, we agree that the fact that patients were taking antihypertensive, antilipidemic and hypoglycemic drugs had a significant impact on the analysis of these results. However, in reality, discontinuing these drugs before the initiation of osilodrostat treatment was impossible due to patient safety. Therefore, in the original manuscript, we have included information in the text regarding the possibility of reducing or withdrawing these drugs in patients.

Reviewer 2 Report

Comments and Suggestions for Authors

This is a small case series

Some language editing needed to make the text "flow" more easily

Table 1: to this reviewer percentages in a case series with n=6 seem superfluous

Comments on the Quality of English Language

Please see aboce

Author Response

„This is a small case series”

We agree that this is a small case series; however, considering how rare Cushing's disease is and that osilodrostat is a relatively new form of treatment for these patients, we believe that it represents a long experience of the therapy.

„Table 1: to this reviewer percentages in a case series with n=6 seem superfluous”

We agree with this comment; it has been considered, and the table has been corrected.

Reviewer 3 Report

Comments and Suggestions for Authors

ID: biomedicines-2725507

Cushing’s disease: long-term efficacy and safety of osilodrostat in Polish group of patients with persistent hypercortisolemia in a single center. by Dzialach L, et al.

To the Authors:

General comments:

The authors investigated real-world evidence regarding the long-term treatment (156 weeks) clinical and biochemical effect of osilodrostat in six patients with Cushing's disease at a single center in Poland.  They found that osilodrostat is effective for disease control and improvement from baseline in most metabolic and cardiovascular parameters.  The theme of this study was considered attractive, and the study was well structured.  I recommend the publication of this manuscript; however, several points should be addressed to improve the manuscript.

Specific comments:

1. The authors should add the figures showing the levels of total serum cortisol and adrenocorticotropic hormone (ACTH) according to osilodrostat dose during the study observation for every patient.  Also, the circulating levels of ACTH should be analyzed in the plasma, not in the serum.  Please clarify this point.

2. The table on page 11 in chapter 3.2, "Effect on metabolic, cardiovascular and liver parameters," should be Table 3, not Table 2.

3. It is considered beneficial to statistically analyze the significance of changes in each parameter in a table titled "Metabolic, cardiovascular and liver parameters at baseline and their changes from baseline at checkpoints during observation" on page 11.

4. The authors should address the combined usage of oral hydrocortisone replacement in addition to osilodrostat administrasion.

Author Response

“The authors should add the figures showing the levels of total serum cortisol and adrenocorticotropic hormone (ACTH) according to osilodrostat dose during the study observation for every patient.  Also, the circulating levels of ACTH should be analyzed in the plasma, not in the serum.  Please clarify this point.”

We considered this comment and presented the change in ACTH concentrations over time for each patient and the median ACTH for all patients in separate graphs. However, we believe the addition graphs of total cortisol serum may be too much, especially since this median did not change significantly throughout the study. In the text, we also included information that ACTH was determined in plasma.

„The table on page 11 in chapter 3.2, "Effect on metabolic, cardiovascular and liver parameters," should be Table 3, not Table 2.”

Thank you for pointing out this error, it has been corrected.

“It is considered beneficial to statistically analyze the significance of changes in each parameter in a table titled "Metabolic, cardiovascular and liver parameters at baseline and their changes from baseline at checkpoints during observation" on page 11.”

Thank you very much for this comment, but we (along with our statistician) believe that more complex statistical inference for such a small group of patients is not entirely justified, mainly since the results of the parameters were influenced by the fact that the patients were taking hypotensive or hypoglycemic drugs. Therefore, it seems to us that it is better to present only Δ% compared to the baseline in the table with an explanation in the text of changes regarding the possibility of reducing or withdrawing specific drugs from treatment in certain patients.

“The authors should address the combined usage of oral hydrocortisone replacement in addition to osilodrostat administrasion.”

We have included additional information regarding this strategy in the text and indicated the possibility of considering the "block and replace" treatment regimen.

Round 2

Reviewer 1 Report

Comments and Suggestions for Authors

the responsed to the comments was satisfactory

Reviewer 3 Report

Comments and Suggestions for Authors

ID: biomedicines-2725507

Cushing’s disease: long-term efficacy and safety of osilodrostat in Polish group of patients with persistent hypercortisolemia in a single center. by Dzialach L, et al.

To the Authors:

General comments:

It is considered that the authors successfully revised the manuscript according to the comments.